# Sex Hormones as Key Modulators of the Immune Response in Multiple Sclerosis: A Review

**DOI:** 10.3390/biomedicines10123107

**Published:** 2022-12-01

**Authors:** Federica Murgia, Florianna Giagnoni, Lorena Lorefice, Paola Caria, Tinuccia Dettori, Maurizio N. D’Alterio, Stefano Angioni, Aran J. Hendren, Pierluigi Caboni, Monica Pibiri, Giovanni Monni, Eleonora Cocco, Luigi Atzori

**Affiliations:** 1Clinical Metabolomics Unit, Department of Biomedical Sciences, University of Cagliari, 09124 Cagliari, Italy; 2Multiple Sclerosis Regional Center, ASSL Cagliari, ATS Sardinia, 09126 Cagliari, Italy; 3Department of Biomedical Sciences, Section of Biochemistry, Biology, and Genetics, University of Cagliari, Cittadella Universitaria, 09124 Cagliari, Italy; 4Division of Gynecology and Obstetrics, Department of Surgical Sciences, University of Cagliari, 09124 Cagliari, Italy; 5Sussex Neuroscience, University of Sussex, Brighton BN1 9QG, UK; 6Department of Life and Environmental Sciences, University of Cagliari, 09124 Cagliari, Italy; 7Department of Biomedical Sciences, University of Cagliari, 09124 Cagliari, Italy; 8Department of Obstetrics and Gynecology, Prenatal and Preimplantation Genetic Diagnosis, Fetal Therapy, Microcitemico Pediatric Hospital “A. Cao”, 09121 Cagliari, Italy; 9Multiple Sclerosis Regional Center, ASSL Cagliari, ATS Sardinia, Department of Medical Sciences and Public Health, University of Cagliari, 09126 Cagliari, Italy

**Keywords:** multiple sclerosis, sex hormones, gender differences, pregnancy, puerperium

## Abstract

Background: A variety of autoimmune diseases, including MS, amplify sex-based physiological differences in immunological responsiveness. Female MS patients experience pathophysiological changes during reproductive phases (pregnancy and menopause). Sex hormones can act on immune cells, potentially enabling them to modify MS risk, activity, and progression, and to play a role in treatment. Methods: Scientific papers (published between 1998 and 2021) were selected through PubMed, Google Scholar, and Web of Science literature repositories. The search was limited to publications analyzing the hormonal profile of male and female MS patients during different life phases, in particular focusing on sex hormone treatment. Results: Both men and women with MS have lower testosterone levels compared to healthy controls. The levels of estrogens and progesterone increase during pregnancy and then rapidly decrease after delivery, possibly mediating an immune-stabilizing process. The literature examined herein evidences the neuroprotective effect of testosterone and estrogens in MS, supporting further examinations of their potential therapeutic uses. Conclusions: A correlation has been identified between sex hormones and MS clinical activity. The combination of disease-modifying therapies with estrogen or estrogen plus a progestin receptor modulator promoting myelin repair might represent an important strategy for MS treatment in the future.

## 1. Introduction

Multiple sclerosis (MS) is a chronic autoimmune disease characterized by demyelinating lesions of the central nervous system (CNS) [1] and represents a major cause of neurological morbidity in young adults [2]. The pathogenesis of MS is characterized by the loss of self-tolerance toward myelin and other CNS antigens, which often leads to persistent activation of autoreactive T lymphocytes [3]. Pathological conditions, especially autoimmune diseases such as MS, often heighten sex-based physiological differences with respect to immunological responsiveness [4]. As a result, the incidence of MS is higher in women than in men, with a ratio of 2–3:1, suggesting a biological sex-related susceptibility [5]. In effect, sex influences the immune response toward both self- and non-self-antigens [4]. Compared to men, women develop stronger innate and adaptive immune responses [6] and have also been found to be more prone to autoimmune diseases [4,7]. On the other hand, men have higher mortality related to infectious diseases and a higher risk of developing fatal cancers [4] (Figure 1A).

The higher female responsiveness to various immunogenic stimuli is also evidenced in the reaction to specific myelin autoantigens in MS [8]. Sex hormones, such as testosterone, estrogens, and progesterone, can influence the immune system (IS) due to the hormone receptors present on immune cells, potentially enabling them to modify MS risk, activity, and progression [5], and to play a role in disease treatment.

On these bases, we aimed to conduct a review of the literature regarding the relationship between sex hormones and the clinical activity of MS, to understand the influence of hormonal alterations. Since currently available therapies reduce inflammation but do not slow the rate of demyelination and neurodegeneration typifying MS, we have evaluated the possible clinical use of one or more sex hormones to treat neurodegeneration in MS patients.

## 2. Materials and Methods

The scientific papers were selected through PubMed (MEDLINE), Google Scholar, and Web of Science repositories using the following keywords: “gender”, “testosterone”, “estrogen”, “progesterone”, “sex”, “pregnancy”, “menopause”, and “multiple sclerosis”. The selected papers were downloaded, after which duplicated works were deleted and the papers not fulfilling the inclusion criteria were removed.

The search was limited to papers published between the years 1998 and 2021, analyzing the effect of the hormonal changes of male and female patients affected by MS and others focusing on sex hormone treatment in MS patients. All studies were realized on humans and published in English, while systematic reviews, meta-analyses, and studies on animal models were excluded.

At first, 84 scientific papers were selected, then 54 of them were excluded since their abstracts did not satisfy the inclusion criteria. Only 18 of the remaining 30 papers have been considered relevant to the aims of this review, due to a lack of focus on MS or early interruption of the study on the part of the other 12 papers.

## 3. Results

The included studies are summarized in Table 1, outlining the data classified by theme, author, title, publication year, sample size, study design, control used, and the results. The identification of a specific control type or study design was not feasible for all studies. Figure 2 represents the summary of the influence of the hormonal changes on the prevalence, relapses, and progression of the MS during the different life phases.

### 3.1. Gender and Multiple Sclerosis

MS risk is three times lower in men than in women. Nevertheless, men have a larger risk of developing severe forms of the disease [26] and have worse long-term prognoses due to the earlier attainment of severe disability [27]. Interestingly, the disease progresses similarly in both sexes when diagnosed after 50 years of age, though it progresses faster in men when diagnosed around the age of 30 [5]. When compared to men, women experience an earlier onset of MS and develop greater numbers of lesions detectable via magnetic nuclear resonance imaging [5,28]. Nevertheless, male sex is an important predictive factor for MS-associated cognitive decline [22] as MS in males is characterized by higher cerebellar involvement, greater cerebral atrophy, and cerebral dysfunction [29]. In addition, males are more frequently affected by primary progressive multiple sclerosis (PPMS) [30] and the recovery after a relapse is weaker when compared with women [31]. Such evidence posits endocrine, neurobiological, and immunological factors as responsible for the aforementioned differences.

### 3.2. Clinical Trials with the Use of Testosterone

Both men and women with MS have lower testosterone levels when compared to healthy age-matched subjects [11]. In a study enrolling 96 MS male subjects, 40% of them showed low testosterone levels; moreover, lower testosterone levels correlated with higher EDSS values (Expanded Disability Status Scale) [11].

Three studies analyzing testosterone as a possible therapeutic agent identified a direct neuroprotective effect [9,12]. In the first phase 2 study, 10 male patients with relapsing-remitting MS (RR-MS) were followed for 6 months preceding testosterone treatment, and for 12 months after [9]. Blood samples were taken periodically before and during the treatment, as well as the PASAT score (Paced Auditory Serial Addition task), magnetic resonance images (MRI), and measurements of cerebral volume. After 12 months of treatment, a remarkable improvement of the PASAT score was observed, with a 67% reduction of cerebral volume loss occurring in the final months of therapy when compared to the pre-treatment period [9].

A later study evaluating the effects of testosterone treatment on grey matter volume in MS patients [12] showed that testosterone treatment prevented grey matter loss, resulting in a significant net increase in the right frontal cortical volume [12].

A further study examined blood samples from the same patients, in which the PBMCs (peripheral blood mononuclear cells) subpopulation, as well as cytokine and growth factor production, were quantified [10]. The obtained data suggested that testosterone treatment induced a shift in the peripheral lymphocytic composition, resulting in a reduction of CD4+ T lymphocytes and an increase of natural killer (NK) cells. Furthermore, IL-2 production from PBMCs was significantly reduced, while transforming growth factor beta 1 (TGF-β1) production increased. PBMCs also produced a higher quantity of brain-derived neurotrophic factor (BDNF) and PDGF (platelet-derived growth factor) after testosterone treatment. These results suggest an immunomodulatory and neuroprotective effect of testosterone treatment in MS patients [10]. Currently, the TOTEM RR-MS trial (Testosterone Treatment on neuroprotection and Myelin Repair in RR-MS) is ongoing, to evaluate the remyelinating and neuroprotective effects of testosterone in MS patients [13]. This phase 2, multicentric, randomized, double-blind French study aims to prevent MS progression in male patients with low testosterone levels through co-administration of testosterone and Natalizumab.

### 3.3. Women’s Life Reproductive Phases and MS

#### 3.3.1. Puberty

Puberty is a period of intensive hormonal change, and thus represents a significant milestone in MS. The onset of MS occurs in childhood in about 5% of cases, though pre-pubertal cases are extremely rare (0.2–0.7%) [32]. Under the age of 9, the sex ratio is 1:1, but female predominance emerges soon after, indicating a potentially crucial effect of the female sex hormone profile on the risk of developing MS during adolescence [33]. No differences were found during puberty between male MS patients and controls [34]. Several retrospective studies investigated the correlation between menarche and the onset of MS in female patients, demonstrating that an earlier menarcheal age was associated with an increased risk for adult-onset MS [35] and an earlier age of MS onset [34,36]. Puberty also affects the course of MS: relapse rates increase during the peri-menarcheal period, while an association exists between older menarcheal age and a reduced risk of reaching a high EDSS score in progressive MS [36,37].

#### 3.3.2. Pregnancy and Post-Partum

Pregnancy is protective against autoimmune disorders, including MS, rheumatoid arthritis, and psoriasis, though not against others, such as systemic lupus erythematosus [14,15]. This observation suggests that the immune alterations taking place during pregnancy induce a protective immune shift in cases of a cell-mediated disease, but not in antibody-mediated disease [27]. Gestation is considered an immune-tolerant condition, where the maternal IS must adapt to the allogenic fetal tissues [14,38]. The fetus–placental unity secretes cytokines, modulating maternal cellular immunity [39] and promoting a strong Th2 response to decrease the risk of miscarriage [15]. In general, during pregnancy, there is an increase in Th2-mediated activity and a reduction of Th1/Th17-mediated activity, which correlate with the worsening of Th2-mediated allergic diseases and the improvement of Th1/Th17-mediated autoimmune diseases [40], respectively.

Studies focusing on MS have shown a 70% reduction of relapses in the third trimester of pregnancy when compared to the clinical situation before pregnancy [14,15,41], a remarkable disparity considering that current MS therapies induce a 40–70% reduction in relapses [5].

This was initially identified in a multinational study conducted by Confavreux et al. [20] on 254 MS women in the year before conception, during pregnancy, and in the post-partum period. In these patients, the relapse rate passed from 0.7 in the pre-pregnancy period to 0.2 per year during gestation, equating to a reduction of more than 70% [20]. The same study showed a relapse frequency in the first three months after childbirth three times greater than that before pregnancy. Moreover, in a recent retrospective observational study, 210 MS women (194 RR-MS) [23] submitted to RMI evaluations on grey and white matters and total cerebral volume estimation, and an increased relapse frequency was observed in the year following the childbirth compared to the year preceding conception [23].

The levels of many hormones change during pregnancy, such as estriol and progesterone [42], potentially playing an immune-stabilizing role.

Progesterone and estrogen concentrations progressively increase during pregnancy, with a peak in the third trimester, then rapidly decrease after delivery. This makes their temporal profile compatible with the protection from MS relapses seen during pregnancy. It is generally assumed that the estrogens’ protective role relies on the induction of anti-inflammatory cytokines’ production and Treg cell proliferation [43]. Correspondingly, during pregnancy, the levels of regulatory T cells (Treg) can increase either in the maternal circulation and in the placenta to suppress the allogenic response against the fetus.

On the other hand, the post-partum period, with its hormonal changes and its increased relapse rate, is associated with an immunological shift toward Th1 responses and estrangement from Th2 responses. Many studies have shown a pathophysiological rebound in the first 3–6 months after delivery. According to the PRIMS study [21], relapse frequency increased from 0.2 in later pregnancy to 1.2 in the first 3 months after delivery [21]. In patients followed for two years, three indices were found to significantly correlate with the risk of a post-partum relapse: (1) an increased relapse rate in the year before pregnancy, (2) an increased relapse rate during pregnancy, and (3) a higher EDSS score [21].

The increased disease activity after childbirth seems to be correlated to the sudden post-partum reduction of estrogens, leading to the loss of the immunosuppressive condition characterizing pregnancy and the elevated pro-inflammatory cytokine concentration occurring in the final phases of pregnancy [22].

#### 3.3.3. Menopause

While the age at MS onset is typically around 20–30, the median age of patients with clinically active MS is usually higher, closer to menopause [44]. During this period, there is a loss of follicular ovarian function matched by a decline in estradiol production [45]. A decline in physiological estrogen levels during menopause has been associated with immunological changes comparable to those observed in MS. These changes include increased production of pro-inflammatory cytokines and decreased secretion of anti-inflammatory cytokines [46].

It has been suggested that low estrogen levels, as observed in menopause, might lead to a relapse in female MS patients [47]. A Swedish study showed that 39% of women affected by MS reported increased symptoms after menopause [24], even though a direct correlation between menopause and relapse was not confirmed by later studies [25,46].

Estrogen deficiency in menopausal women has been associated with increased levels of Th17 lymphocytes and IL-17 production [48].

Although some studies suggested that female MS sufferers experience accelerated disability progression after menopause, current data are not solid enough to draw firm conclusions [49]. Several confounding factors take part in this complex scenario: for example, older MS patients have many years of accumulating damage in their CNS. Moreover, women with MS might experience a faster progression of disability after menopause, showing a conversion to the SPMS form (common at this age) which overlaps with the neuroinflammation and neurodegeneration occurring during physiological ageing [50], where an important role is played by immune senescence, affecting both the innate and adaptive IS [51]. Additionally, the increased production of pro-inflammatory cytokines may have a role [51].

### 3.4. Clinical Trials with the Use of Estrogens

The current literature supports the use of estrogens as therapeutic agents in MS. In the last twenty years, many clinical trials have been conducted on estriol as a possible MS therapeutic agent. In 2002, a phase 2 single-arm study enrolled 10 female MS patients (6 RR-MS and 4 secondary progressive MS, SPMS). Each was treated daily with estriol for six months, while clinical evaluation and lab tests were undertaken monthly. Cerebral magnetic resonance imaging was performed before, during, and after the hormone treatment. The magnetic resonance imaging (MRI) registered significant reductions in the number and volume of lesions in the patients affected by RR-MS [15], but when the estriol treatment ceased, the enhancing lesions returned, reverting to the pre-treatment situation [14].

Subsequently, the trial was extended to the 6 RR-MS patients who underwent treatment with estriol and progesterone for 4 months. During this period, the volume and the number of enhancing lesions decreased again compared to the original situation [14].

A study performed on the same blood samples evidenced a partial shift in the PBMC population from Th1- to Th2-dominant. In addition, increased production of IL-5 and IL-10, and a reduction of tumor necrosis factor alpha (TNF-α), were identified in RR-MS patients, correlating with the RMN lesion and suggesting direct estriol anti-inflammatory effects in the brain [15].

One notable phase 2 multicentric, double-blinded study recruited 164 female PRMS sufferers, who were randomly separated into two groups and treated with glatiramer acetate (GA) plus oral estriol or a placebo [41]. Treatment with estriol decreased relapse frequency in the RR-MS [41] group compared to the placebo group, while improving cognitive function and reducing fatigue. Moreover, higher estriol levels corresponded to a lower number of relapses and gadolinium-enhancing lesions, and a positive correlation was observed between the PASAT scores and the grey matter volume upon estriol intake [41]. These data were used in a later study enrolling RR-MS patients treated with GA plus estriol or GA with placebo. As evaluated by voxel-based morphometry VBM and PASAT scores, the grey matter areas spared during estriol treatment were associated with improved cognitive performance [17].

Another study showed that matrix metalloproteinase-9 levels, which mediate the transmigration of T lymphocytes and monocytes in the CNS and are known to increase in different inflammatory CNS diseases, including MS [52], decreased in female RR-MS patients treated with high estriol doses. This reduction was matched by a corresponding reduction of RMN-enhancing lesions [16]. 

Estradiol was also evaluated as a potentially protective factor against MS [18]. The trial involved RR-MS patients who were randomly divided into three groups to receive a 96-week course of interferon beta 1a (IFN-β1) plus a placebo or 20 or 40 µg of ethinyl estradiol. The level of active inflammatory nuclear magnetic resonance (NMR) lesions decreased by 26% in the group treated with 40 µg of ethinyl estradiol compared to the placebo group, while the group that took 20 µg of the hormone showed an intermediate but not significant reduction [18].

A subsequent study performed on the same cohort of patients showed that the 40 µg dosage of ethinyl estradiol, but not the 20 µg dosage, was associated with better cognitive performances compared to the placebo [19].

These results highlight the importance of the type and the dosage of estrogen when considering it as a possible therapeutic agent.

## 4. Discussion

Sex hormones undoubtedly play a modulatory key role in immune function, with their effects varying depending on hormone concentration, target cell type, and cell receptor subtype [8]. These are key protagonists in the pathogenesis of MS, where the loss of self-tolerance toward myelin and other CNS antigens leads to persistent activation of autoreactive Th1 lymphocytes [3]. Subsequently, by encountering auto-antigens inside the cerebral parenchyma, this loss of self-tolerance triggers an inflammatory cascade, inducing massive myelin damage [1]. On the other hand, Th2 lymphocytes play a counter-regulatory role, limiting Th1-mediated damage [53]. Sex hormones also influence the reciprocal inhibition between these two responses [42].

Androgens can modulate immune function by altering gene transcription through many mechanisms, including DNA binding patterns [54]. Similarly, androgen receptors (AR) are expressed in various organs, as well as the cells of the IS, indicating their involvement in immunity. Indeed, thymocytes, epithelial thymic cells, peripheral T lymphocytes, bone marrow stromal cells, and B lymphocytes’ precursors express intracellular ARs [55,56,57,58]. Moreover, ARs are expressed by both myeloid precursor and myeloid differentiated cells [59]. Therefore, testosterone can modulate the development and function of both lymphoid and myeloid branches of the IS (Figure 1B). It has been proposed that testosterone could have neuroprotective effects in MS [60], by crossing the blood–brain barrier, protecting neuronal cells from oxidative stress [61], and inducing the expression of neurotrophic factors such as BDNF (brain-derived neurotrophic factor) [62].

Circulating estrogen levels (estrone, E1; 17β-estradiol, E2; estriol, E3) change throughout a woman’s life, from childhood to menopause. They can influence the IS and CNS activity [8] through binding to nuclear estrogen receptors (ERs), ER-α and ER-β, and altering gene transcription and non-genomic signaling pathways [8].

Changes in circulating estrogen levels can affect both progenitors and mature cells of the innate and adaptive IS [8], regulating their proliferation and their specific biological functions [6] (Figure 1B). The action of estrogens on the IS is well-studied in physiological (menstrual cycle, pregnancy) and pathological immune conditions [63], including MS. During MS, ER-α and ER-β activation exerts an anti-inflammatory effect via modulation of microglia and mediates neuroprotection through effects on CNS macrophages. Both ER-α- and ER-β-mediated signals reduce demyelination, axonal loss, and neuronal disease in experimental autoimmune encephalomyelitis (EAE). Moreover, ER activation induces oligodendrocyte maturation and potentiates remyelination [64] through expression in both immature and mature oligodendrocytes. Since astrocytes and microglia also express ERs, microgliosis and astrocytosis may be included in the profile of hormonally mediated protective effects against MS [65].

As for physiological conditions, low E2 concentrations during the pre-ovulatory phase of the menstrual cycle induce IFN-γ production from Th lymphocytes [7], promoting cell-mediated immunity. On the other hand, high E2 doses during pregnancy induce TGF-β and IL-10 expression in immune cells [7], promoting immunotolerance by significantly reducing the IFN-γ/IL-10 ratio [64]. Furthermore, estradiol exerts an anti-inflammatory effect by determining the proliferation of Treg lymphocytes [66] and decreasing the levels of Th17 lymphocytes, which are associated with the pathogenesis of various autoimmune diseases, including MS [67].

In summary, estrogens have a biphasic, dose-dependent effect, and low doses facilitate while high doses inhibit the immune response [8]. This seems to influence the clinical course of MS, where high estrogen levels reduce pro-inflammatory molecule production and increase the stimulation of Th2-mediated anti-inflammatory pathways [5]. However, high doses of estrogens mimicking the hematic concentration of third-trimester pregnancy are not well-tolerated, due to their high affinity for breast and uterine ER-α, which can induce tumorigenesis. These data underline the importance of the type and the dose of estrogen to be used in the MS treatment.

Estrogens also affect B lymphocyte maturation, differentiation, activity, and survival [68]. It has been proven that estrogens increase plasma cell numbers and promote IL-10 secretion from B regulatory lymphocytes (Breg) that negatively regulate the T cell response [64]. B lymphocytes contribute to the pathogenesis of MS by acting as antigen-presenting cells and by producing anti-myelin antibodies and cytokines [69]. Like Treg lymphocytes, Breg lymphocytes also proliferate during pregnancy [70].

Progesterone has a broad spectrum of cerebral actions: it influences sexual behavior and reproduction, myelination, and neuroprotection, as well as CNS development and differentiation [71] (Figure 1B). Circulating progesterone can easily cross the blood–brain barrier and spread in the nervous tissues. In addition, it can be locally synthesized in the CNS by neurons and glial cells. Its activity stems from its interaction with progestin receptors (PR) [72], densely expressed on immune cells [4] (Figure 1B). In general, progesterone switches the immune response from pro-inflammatory to anti-inflammatory, promoting Treg lymphocytes’ differentiation and downregulating IFN-γ production from NK cells. In addition, progesterone promotes Th2 differentiation in vitro, and reduces nitric oxide production in macrophages [72] (Figure 1B).

During pregnancy, the combined action of estrogens and progesterone can realize and maintain an immune-tolerant environment [73]. On the other hand, MS symptoms worsen during the premenstrual period when progesterone levels decrease [74].

During MS, progesterone modulates the remyelination process [65] by increasing the rate of myelin synthesis and upregulating myelination via action on neurons [75]. It is still unclear whether a beneficial increase in myelin synthesis results from direct actions on oligodendrocytes or indirect effects via other cells such as astrocytes or macrophages [65]. Despite this, there is abundant evidence that it is a crucial regulator of myelination and myelin development and repair after CNS diseases [65].

## 5. Conclusions

Based on the reviewed scientific studies, a correlation between sex hormones and the clinical activity of MS has been found. The most convincing case is exemplified by women with MS during the third trimester of pregnancy, where a positive correlation exists between attenuated clinical activity and estrogen and progesterone peaks. This explains why various trials have been carried out to recreate the immunologically beneficial “pregnancy status”. The estrogen that has been paid the greatest attention is estriol due to its typical fluctuations during pregnancy and its current widespread therapeutic application, meaning there is no need to examine its toxicity in phase 1 studies. Indeed, it is considered the safest of the three known estrogens (paying attention to the suitable dose).

Novel MS therapies aimed at reducing inflammation mainly focus on the autoimmune nature of the disease. Even though anti-inflammatory and immunomodulatory treatments can decrease the relapse frequency, they scarcely prevent neurodegeneration, axonal loss, and clinical disability. Therefore, current research focuses on agents that can inhibit demyelination, stimulate remyelination, and/or prevent axonal damage, thus avoiding the progression of clinical disability. The combination of an anti-inflammatory agent with estrogen or the combination of estrogen with a progestin receptor modulator promoting myelin repair might represent an important strategy for MS treatment in the future.

At present, only two phase 2 studies with the use of estriol or testosterone have demonstrated positive results. They might represent therapeutic agents with low-cost and few collateral effects, and their inclusion in phase 3 studies is highly recommended. Moreover, as only a few studies have investigated the correlation between the worsening of MS progression and menopause, this could represent an important aspect to explore to deepen the pathophysiological mechanisms behind MS.

## Figures and Tables

**Figure 1 biomedicines-10-03107-f001:**
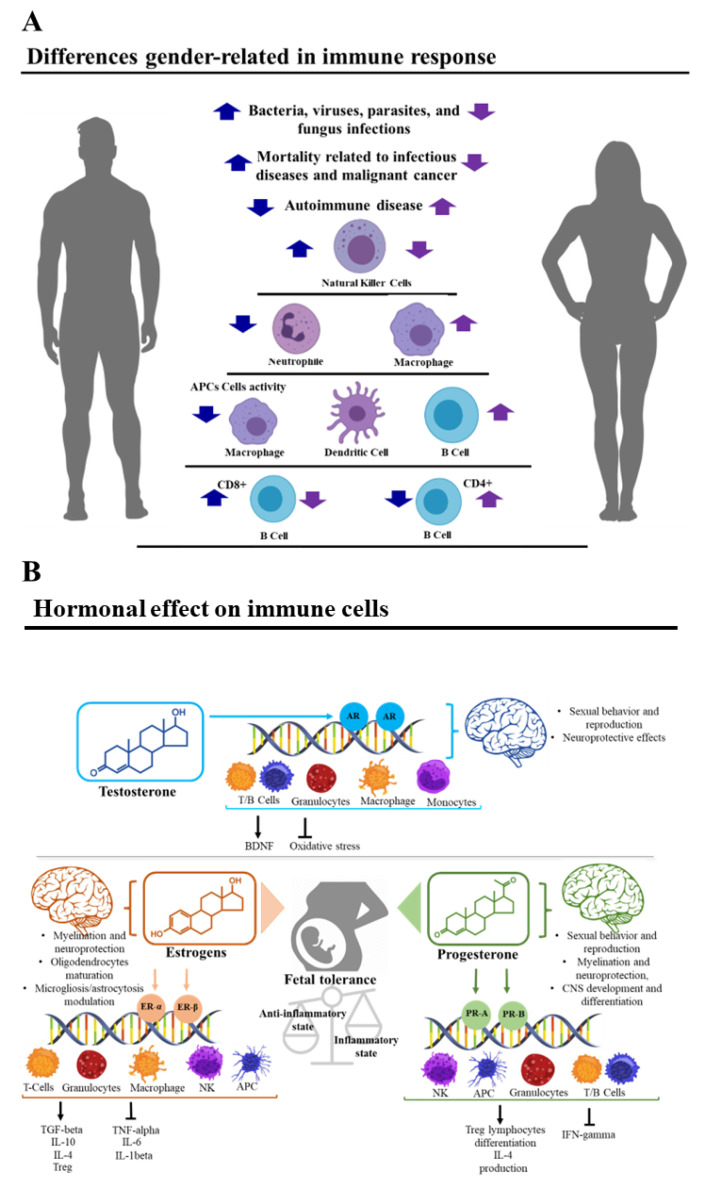
(**A**) Main differences in the immune response between men and women. Blue arrows represent the trend in the men while violet arrows represent the trend in the women. Innate immune cell number and activity differ between males and females. Males present a higher number of natural killer cells compared to females, while phagocytic activity of neutrophils and macrophages is higher in females [4]. Additionally, antigen-presenting cells (APCs) are more efficient in females than males [4]. Sex hormones and an XY karyotype also influence adaptive immunity. Differences between the two sexes are present in B lymphocytes, CD4+, and CD8+ T lymphocytes. For example, females present with a higher CD4+ T lymphocytes count and a higher CD4/CD8 ratio compared to males of the same age, while males exhibit a higher CD8+ T lymphocytes count [4]. Women show higher numbers of activated CD4+ and CD8+ T lymphocytes and a higher number of proliferating T lymphocytes compared to men. Furthermore, women show higher levels of immunoglobulins and higher numbers of B lymphocytes compared to men [4]. (**B**) Some examples of immune modulation by sex hormones. Classic hormone-receptor transduction is a multi-step process where receptor dimerization, ligand binding, cofactor interaction, and DNA binding take place. Once the ligand is bound, the receptor-ligand complex migrates into the nucleus to specific DNA binding sites, regulating transcription. Androgen receptors (AR) are expressed in various organs, as well as the cells of the immune system, indicating their involvement in immunity. Estrogenic receptors (ERs) regulate pathways of both innate and adaptive immune systems, while concurrently regulating immune cell development and function, explaining, in part, the differences between the innate pathway responses in the two sexes. Changes of circulating estrogen levels can affect progenitors and mature cells of the innate and adaptive immune system, regulating the number and the specific biological functions of the cells (neutrophils chemiotaxis, infiltration, and cytokine production, macrophage chemiotaxis, phagocytic activity and cytokine production, natural killer (NK) cell cytotoxicity, dendritic cell differentiation, and cytokine expression).

**Figure 2 biomedicines-10-03107-f002:**
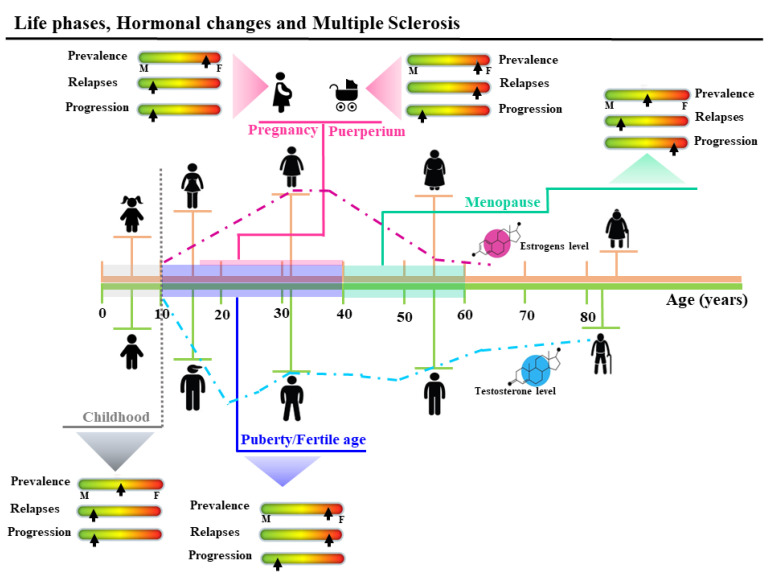
Summary of the effects of the hormonal changes on the prevalence, relapse, and progression of the multiple sclerosis (MS) during the different life phases. Before puberty (age 10 years), MS prevalence is similar between male and females. After menarche, in line with the increase of the estrogens level, prevalence in girls is three times higher. During the fertile age, female predominance remains and relapses are shown in both sexes, but during pregnancy, a significant decrease in female relapse rates (about 70%) is observed, especially in the third trimester. After delivery during the puerperium, relapse rates increase to three times higher than pre-pregnancy levels. Worsening of symptoms and progression of MS have been shown in menopause.

**Table 1 biomedicines-10-03107-t001:** Summary of the eligible studies discussed in the review.

Authors	Title	Year	Sample Size	Study Group	Group of Control	Result
**Testosterone**						
Sicotte N.L. et al. [9]	Testosterone treatment in multiple sclerosis	2007	10 men with RR-MS	10 men with RR-MS treated daily with gel containing 100 mg of testosterone.	Each patient served as his own control	Treatment with testosterone gel for 12 months was associated with improved cognition and slowing of brain atrophy
Gold S.M. et al. [10]	Immune modulation and increased neurotrophic factor production in Multiple Sclerosis patients treated with testosterone	2008	Blood samples from 10 men with RR-MS	10 men with RR-MS treated daily with gel containing 100 mg of testosterone.	Each patient served as his own control	Increased production of BDNF and PDGF-BB suggests a potential neuroprotective effect.
Bove R. et al. [11]	Low testosterone is associated with disability in men with multiple sclerosis	2014	96 men with RR-MS	96 men with RR-MS	-	Lower testosterone levels were correlated with higher EDSS.
Kurth F. et al. [12]	Neuroprotective effects of testosterone treatment in men with multiple sclerosis	2014	10 men with RR-MS	10 men with RR-MS treated daily with gel containing 100 mg of testosterone.	Each patient served as his own control	These observations may reflect the potential of testosterone treatment to reverse gray matter atrophy associated with MS
Metzger P.K. et al. [13]	The TOTEM RR-MS (testosterone treatment on neuroprotection and myelin repair in RR-MS) trial	2020	40 testosterone-deficent men with RR-MS	Patients treated with Natalizumab + testosterone	Patients treated with Natalizumab + placebo	Study still ongoing
**Estriol**						
Sicotte L. et al. [14]	Treatment of multiple sclerosis with the pregnancy hormone estriol	2002	10 female patients with clinically definite MS (6 RR/4SP)	All the patients were treated with estriol (8 mg)	Each patient served as her own control	The total volume and number of enhancing lesions decreased during the treatment
Soldan S. et al. [15]	Immune modulation in multiple sclerosis patients treated with the pregnancy hormone estriol	2003	10 female patients with clinically definite MS (6 RR/4 SP)	All the patients were treated with estriol (8 mg)	Each patient served as her own control	Increased production of IL-5 and IL-10 and decreased TNF-alpha during estriol treatment
Gold S.M. et al. [16]	Estrogen treatment decreases matrix metalloproteinase (MMP)-9 in autoimmune demyelinating disease through estrogen receptor alpha (ERα)	2009	3 premenopausal female patients with RR-MS	All the patients were treated with estriol (8 mg/die)	Each patient served as her own control	RR-MS patients treated with pregnancy levels of estriol showed a decrease in MMP-9 levels and activity and in enhancing lesions on MRI.
MacKenzie-Graham A. et al. [17]	Estriol-mediated neuroprotection in multiple sclerosis localized by voxel-based morphometry	2018	164 patients	83 patients treated with glatiramer acetate + estriol (8 mg/die)	81 patients treated with glatirameracetate + placebo	Voxel-based morphometry analysis of gray matter revealed that it was significantly preserved in estriol + GA-treated subjects, but not in placebo + GA-treated subjects.
**Estradiol**						
Pozzilli C. et al. [18]	Oral contraceptives combined with interferon-β in multiple sclerosis	2015	150 women with RR-MS	Patients treated with IFN-β-1a + ethinylestradiol (20 mg) and desogestrel (150 mg) Patients treated with IFN-β-1a + ethinylestradiol (40 mg) and desogestrel (125 mg)	Patients treated with IFN-β-1a only	The anti-inflammatory effect of treatment, as measured by MRI activity, was more pronounced in patients receiving high-dose estrogens than in those receiving IFN-β alone.
De Giglio L. et al. [19]	Effect on cognition of estroprogestins combined with interferon-β in multiple sclerosis: analysis of secondary outcomes from a randomized controlled trial	2016	150 women with RR-MS	Patients treated with IFN-β-1a + ethinylestradiol (20 mg) and desogestrel (150 mg) Patients treated with IFN-β-1a + ethinylestradiol (40 mg) and desogestrel (125 mg)	Patientstreated with IFN-b-1a only	A significant proportion of patients in the group treated with high-dose estrogens improved their cognitive status compared with those treated with IFN-β alone.
**Pregnancy and post-partum**						
Confavreux C. et al. [20]	Rate of pregnancy- related relapse in multiple sclerosis	1998	254 women with MS	254 women with MS		Mean rate of relapse per year before pregnancy = 0.7 Mean rate of relapse during the third trimester of pregnancy = 0.2 Mean rate of relapse during the first three months post-partum = 1.2
S.Vukusic et al. [21]	Pregnancy and multiple sclerosis (the PRIMS study): clinical predictors of post-partum relapse	2004	227 women with MS	227 women with MS		Three prognostic factors of high post-partum relapse rate: (1) increased relapse rate in the pre-pregnancy year, (2) an increased relapse rate during pregnancy, (3) higher DSS score at pregnancy onset, significantly correlated with the occurrence of a post-partum relapse
L. Airas et al. [22]	Immunoregulatory factors in multiple sclerosis patients during and after pregnancy: relevance of natural killer cells	2007	42 pregnant (RR-MS)	42 pregnant (RR-MS)		Reduced disease activity during the last trimester was associated with a significant increase in the percentage of circulating CD56-bright natural killer cells
Lorefice L. et al. [23]	Effects of pregnancy and breastfeeding on clinical outcomes and MRI measurements of women with multiple sclerosis: an exploratory real-world cohort study	2021	210 women with MS	210 women with MS		A higher annualized relapse rate in the post-partum year versus the pre-conception year was observed. Pregnancy during MS was associated with a lower EDSS score, while no relationships were reported with MRI measurements.
**Menopause**						
Holmqvist P. et al. [24]	Symptoms of multiple sclerosis in women in relation to sex steroid exposure	2005	128 women with MS	128 women with MS		39% of the women reported worsening of MS symptoms related to menopause, whereas 56% reported no change of symptoms and 5% reported decreased symptoms.
Ladeira F. et al. [25]	The influence of menopause in multiple sclerosis course: a longitudinal cohort study	2018	37 women with MS	37 women with MS		Following menopause, a reduction in the relapse rate was observed, but the disability progression continued at a similar rate, compared to the pre-menopausal period.

## Data Availability

Not applicable.

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
