# Peer review of "Sex Hormones as Key Modulators of the Immune Response in Multiple Sclerosis: A Review"

_biomedicines, 2022, doi:10.3390/biomedicines10123107_

Round 1

Reviewer 1 Report

Ad Introduction, figure 1A - the last level of the figure showing the trend in CD4+ and CD8+ lymphocytes counts among males and females. It is not clear what means B cells on that point; what the authors want to  show.

Talking about organization of the work -  I suggest the information about trials with  the use of testosterone in MS patients (line numbers: 159-170, 179-184) should be presented smilarly to data regarding estrogens. They should be presented in separate paragraph (with separate numbuer), after data how testosterone influences MS course, how it influences immune system.

Ad. Conclusions, I suggest, they shoud be modified. They should be presented more transparently. Beside sentences they sound as conclusions, this part of the manuscript involves informations, they should be indicated rather in discussion section or in results section; /"As for estradiol, high doses mimicking 395 the haematic concentration of third-trimester pregnancy are not well tolerated, due to its 396 high affinity for breast and uterine ERα, which can induce tumorigenesis"/ 

Ad English language: the title of figure 1A, I suggest  - Gender-related diffrences in immune response ; in conclusion section - line number 387 - Based on the examined scientific studies - would be correct.

Author Response

We would like to thank the reviewer for the deep analysis of our review. We strongly believe that his suggestions can improve the manuscript. Hoping to have done a good job, we tried to address the following points.

Ad Introduction, figure 1A - the last level of the figure showing the trend in CD4+ and CD8+ lymphocytes counts among males and females. It is not clear what means B cells on that point; what the authors want to  show.

  1. Thank you for this important observation. We apologize for the misunderstanding.

Talking about organization of the work -  I suggest the information about trials with  the use of testosterone in MS patients (line numbers: 159-170, 179-184) should be presented smilarly to data regarding estrogens. They should be presented in separate paragraph (with separate numbuer), after data how testosterone influences MS course, how it influences immune system.

  1. Thank you for the suggestion. We added the title of the paragraph “Clinical trials with the use of testosterone”.

Ad. Conclusions, I suggest, they should be modified. They should be presented more transparently. Beside sentences they sound as conclusions, this part of the manuscript involves informations, they should be indicated rather in discussion section or in results section; /"As for estradiol, high doses mimicking 395 the haematic concentration of third-trimester pregnancy are not well tolerated, due to its 396 high affinity for breast and uterine ERα, which can induce tumorigenesis"/ 

  1. We have modified the conclusion section following your suggestion.

Ad English language: the title of figure 1A, I suggest - Gender-related diffrences in immune response ; in conclusion section - line number 387 - Based on the examined scientific studies - would be correct.

  1. Done!

Reviewer 2 Report

This manuscript by F. Murgia et al. is a literature review on the role of hormones in MS and the potential for sex hormones modulators as a future MS therapy.   

In general, the manuscript is well written, easy to read and provides a good background knowledge on the role of hormones in MS.

I have the following comments/questions that would help improve this manuscript:

The authors should define all acronyms the first time they appear in the manuscript (BDNF, TGF-beta, NK cells, etc.)

Line 34: Period is missing at the end of the sentence

Line 37: I assume the authors mean “the neuroprotective effect of testosterone….”

Line 37-38: “supporting further examinations …. uses.” remove the “a”

Line 48-49: characterized by “the” loss of

Line 49, 317: encephalitogenic T cells are not constantly activated otherwise there would be no remission in RRMS patients. There is a precarious balance between regulation/remyelination and reactivation/epitope spreading. Please modify.  

Line 56: diseases

Line 56: “However” is the wrong adverb to use here. “On the other hand” or a synonym would make more sense.

Line 58: cancers

Further grammatical errors, typos and suboptimal wording can be found throughout the manuscript. I would advise the authors to review the text carefully and make appropriate corrections. In most cases, the errors to not ampere understanding.

FIGURE 1:

The title in section A should read “Gender-related differences in immune responses”.

The text should read “Bacterial, viral, parasitic and fungal infections”, “… malignant cancers”, “autoimmune diseases” “neutrophils” “macrophages” “dendritic cells” “B cells”

In section B, the title should be “Hormonal effects on immune cells”.

Please make sure you always write the plural of the name of the cells.

The left brain is hiding some of the text.

Line 61-62: innate immune cell number

Line 63: if only the phagocytic activity is different in neutrophils and macrophages, this should be made more obvious in the figure; otherwise, it looks like the difference stated is in numbers.

In addition, the lower section of the section A should be modified. As it stands, it looks like CD8+ B cells and CD4+ B cells are discussed. A T cell cartoon must be included. While CD4+ lymphocytes cannot be mistaken for another cell, some B lymphocytes have been shown to express CD8. I would therefore suggest to the author to include the “T” of T lymphocytes when appropriate in the text to avoid confusion. Normally it would be written like so: CD4+ T lymphocytes (also line 175).  

The scale under fetal tolerance is confusing. At first glance, it seems to show that the left side, estrogen, is anti-inflammatory, while the right side, progesterone, is pro-inflammatory. Since this is not the case, I would suggest modifying the figure to prevent misunderstandings.

The bottom left side: make sure that Treg differentiation and IL-4 secretion by Th2 cells are well separated. As written, it could be understood that Treg cells secrete IL-4.

Page 6: In S. Vukusic et al., perhaps writing “Three prognostic factors of high post-partum relapse rate” instead of “Three indices” would define better the statement.

In L. Airas et al., a dash is needed between CD56 and bright (CD56-bright)

In Figure 2, the bar charts showing the higher prevalence, relapses and progression between males and females during pregnancy and postpartum, is confusing. Are the authors still referring to a higher likelihood between males and pregnant females or is it rather comparing females during and after pregnancy to females without pregnancy? If it is the latter, please modify the graphic.  

Line 239-240: please rephrase

In the menopause section, the authors should include that older MS patients have had many years of accumulating damage in their CNS and the fact that there is a higher prevalence of conversion to SPMS, both of which are confounding factors here.

Line 283,300, 304: please define RMN. Do the authors mean NMR, nuclear magnetic resonance? Line 300, if my understanding is correct, in this context the authors probably mean contrast-enhancing lesions (or gadolinium-enhancing lesions).

Line 303-304: please rephrase for clarity, I believe all patients received interferons.

In the discussion, a good portion on estrogen is repeated from the lengthy legend of figure 1. It would be good to modify this section to include additional information and “discuss” the provided data.

References do not match with the text, it seems like they are shifted down one spot. Please verify.

Final thoughts: The literature chosen for this review has been well selected and the overall results presented are of interest. I was, however, expecting a more in-deep review considering the number of authors on the manuscript. While the authors briefly glance over some aspects of the results presented, there is still a measurable value in this review. Therefore, I would recommend the manuscript for publication if the authors can address my comments above.    

Author Response

This manuscript by F. Murgia et al. is a literature review on the role of hormones in MS and the potential for sex hormones modulators as a future MS therapy.   

In general, the manuscript is well written, easy to read and provides a good background knowledge on the role of hormones in MS.

I have the following comments/questions that would help improve this manuscript:

The authors should define all acronyms the first time they appear in the manuscript (BDNF, TGF-beta, NK cells, etc.)

  1. Thank you, done!

Line 34: Period is missing at the end of the sentence

  1. Done

Line 37: I assume the authors mean “the neuroprotective effect of testosterone….”

  1. yes! We corrected the sentence

Line 37-38: “supporting further examinations …. uses.” remove the “a”

  1. Done!

Line 48-49: characterized by “the” loss of

  1. Done!

Line 49, 317: encephalitogenic T cells are not constantly activated otherwise there would be no remission in RRMS patients. There is a precarious balance between regulation/remyelination and reactivation/epitope spreading. Please modify.  

  1. Thank you for the suggestion. We made the sentence a little more generic.

Line 56: diseases

  1. Done!

Line 56: “However” is the wrong adverb to use here. “On the other hand” or a synonym would make more sense.

  1. Done!

Line 58: cancers

  1. Done!

Further grammatical errors, typos and suboptimal wording can be found throughout the manuscript. I would advise the authors to review the text carefully and make appropriate corrections. In most cases, the errors to not ampere understanding.

FIGURE 1:

The title in section A should read “Gender-related differences in immune responses”.

  1. Done!

The text should read “Bacterial, viral, parasitic and fungal infections”, “… malignant cancers”, “autoimmune diseases” “neutrophils” “macrophages” “dendritic cells” “B cells”

R: Done!

In section B, the title should be “Hormonal effects on immune cells”.

  1. Done!

Please make sure you always write the plural of the name of the cells.

  1. Done!

The left brain is hiding some of the text.

  1. Done!

Line 61-62: innate immune cell number

  1. Done!

Line 63: if only the phagocytic activity is different in neutrophils and macrophages, this should be made more obvious in the figure; otherwise, it looks like the difference stated is in numbers.

  1. We modified the figure 1A

In addition, the lower section of the section A should be modified. As it stands, it looks like CD8+ B cells and CD4+ B cells are discussed. A T cell cartoon must be included. While CD4+ lymphocytes cannot be mistaken for another cell, some B lymphocytes have been shown to express CD8. I would therefore suggest to the author to include the “T” of T lymphocytes when appropriate in the text to avoid confusion. Normally it would be written like so: CD4+ T lymphocytes (also line 175).  

  1. Thank you for this important suggestion. We modified the figure and the text!

The scale under fetal tolerance is confusing. At first glance, it seems to show that the left side, estrogen, is anti-inflammatory, while the right side, progesterone, is pro-inflammatory. Since this is not the case, I would suggest modifying the figure to prevent misunderstandings.

  1. The scale under “fetal tolerance” indicated that during pregnancy predominate the anti-inflammatory state compared to the inflammatory state. We modified the figure made it a little more understandable

The bottom left side: make sure that Treg differentiation and IL-4 secretion by Th2 cells are well separated. As written, it could be understood that Treg cells secrete IL-4.

  1. Done, we added a comma

Page 6: In S. Vukusic et al., perhaps writing “Three prognostic factors of high post-partum relapse rate” instead of “Three indices” would define better the statement.

  1. Done!

In L. Airas et al., a dash is needed between CD56 and bright (CD56-bright)

  1. Done!

In Figure 2, the bar charts showing the higher prevalence, relapses and progression between males and females during pregnancy and postpartum, is confusing. Are the authors still referring to a higher likelihood between males and pregnant females or is it rather comparing females during and after pregnancy to females without pregnancy? If it is the latter, please modify the graphic.  

  1. The pink bar was referred to the woman reproductive age. We modified the figure.

Line 239-240: please rephrase

  1. Done!

In the menopause section, the authors should include that older MS patients have had many years of accumulating damage in their CNS and the fact that there is a higher prevalence of conversion to SPMS, both of which are confounding factors here.

  1. We modified the text following your suggestion.

Line 283,300, 304: please define RMN. Do the authors mean NMR, nuclear magnetic resonance?

  1. Done

Line 300, if my understanding is correct, in this context the authors probably mean contrast-enhancing lesions (or gadolinium-enhancing lesions).

  1. We mean gadolinium-enhancing lesions

Line 303-304: please rephrase for clarity, I believe all patients received interferons.

R: All patients received glatiramer acetate. We specified this point.

In the discussion, a good portion on estrogen is repeated from the lengthy legend of figure 1. It would be good to modify this section to include additional information and “discuss” the provided data.

  1. Thank you for the suggestion. We modified the caption of the figure.

References do not match with the text, it seems like they are shifted down one spot. Please verify.

  1. Done!

Final thoughts: The literature chosen for this review has been well selected and the overall results presented are of interest. I was, however, expecting a more in-deep review considering the number of authors on the manuscript. While the authors briefly glance over some aspects of the results presented, there is still a measurable value in this review. Therefore, I would recommend the manuscript for publication if the authors can address my comments above.  

We really appreciate your deep report which provided several helpful suggestions. Hoping to have done a good job, we tried to address your points.